# Socioeconomic and Governance Factors Disentangle the Relationship between Temperature and Antimicrobial Resistance: A 10-Year Ecological Analysis of European Countries

**DOI:** 10.3390/antibiotics12040777

**Published:** 2023-04-19

**Authors:** Andrea Maugeri, Martina Barchitta, Roberta Magnano San Lio, Antonella Agodi

**Affiliations:** Department of Medical and Surgical Sciences and Advanced Technologies “GF Ingrassia”, University of Catania, Via S. Sofia 87, 95123 Catania, Italy; andrea.maugeri@unict.it (A.M.); martina.barchitta@unict.it (M.B.); robertamagnanosanlio@unict.it (R.M.S.L.)

**Keywords:** government, temperature, drug resistance, economic factors, social factors

## Abstract

Although previous studies showed that warmer temperatures may be associated with increased antimicrobial resistance (AMR) rates, unmeasured factors may explain the observed relationship. We conducted a ten-year ecological analysis to evaluate whether temperature change was associated with AMR across 30 European countries, considering predictors that can determine a geographical gradient. Using four data sources, we created a dataset of: annual temperature change (FAOSTAT database); AMR proportions for ten pathogen–antibiotic combinations (ECDC atlas); consumption of antibiotics for systemic use in the community (ESAC-Net database); population density, gross domestic product (GDP) per capita, and governance indicators (World Bank DataBank). Data were obtained for each country and year (2010–2019) and analyzed through multivariable models. We found evidence of a positive linear association between temperature change and AMR proportion across all countries, years, pathogens, and antibiotics (β = 0.140; 95%CI = 0.039; 0.241; *p* = 0.007), adjusting for the effect of covariates. However, when GDP per capita and the governance index were included in the multivariable model, temperature change was no longer associated with AMR. Instead, the main predictors were antibiotic consumption (β = 0.506; 95%CI = 0.366; 0.646; *p* < 0.001), population density (β = 0.143; 95%CI = 0.116; 0.170; *p* < 0.001), and the governance index (β = −1.043; 95%CI = −1.207; −0.879; *p* < 0.001). Ensuring the appropriate use of antibiotics and improving governance efficiency are the most effective ways of counteracting AMR. It is necessary to conduct further experimental studies and obtain more detailed data to investigate whether climate change affects AMR.

## 1. Introduction

Antimicrobial resistance (AMR) has become a significant threat to global health as a result of higher disease severity, prolonged illness, a substantial number of deaths and disability-adjusted life years (DALYs), and higher healthcare costs due to infections caused by antibiotic-resistant pathogens [1]. This is a major concern, especially in Europe, with many countries reporting high levels of AMR among different bacterial strains [2,3]. On a European and national level, AMR pathogens of greatest public health importance are monitored and specific action plans developed to mitigate the threat [4,5]. Since the late 1990s, the European Antimicrobial Resistance Surveillance Network (EARS-Net) has collected data from national antibiotic resistance surveillance networks across the European Union (EU) and the European Economic Area (EEA), an effort coordinated since 2010 by the ECDC [6].

From that point on, numerous national and international studies consistently estimated that the burden of infections with antibiotic-resistant bacteria exceeds that of other infectious diseases [1,2,7,8,9]. This is at least partially explained by the overall burden of healthcare-associated infections (HAIs), which are often caused by antibiotic-resistant pathogens [2,10,11,12,13]. 

In spite of a slight decline from 2019 to 2020, an increasing trend of infections, attributable deaths, and DALYs due to antibiotic-resistant pathogens has been observed during the last decade [3]. These recent estimates from the EU/EEA show that the annual number of infections with AMR pathogens ranged from approximately 685,000 in 2016 to 866,000 in 2019. Similarly, the annual number of attributable deaths increased from nearly 31,000 in 2016 to 39,000 in 2019 [3]. The largest burden was attributable to *Escherichia coli* and *Klebsiella pneumoniae* resistant to third-generation cephalosporin and to methicillin-resistant *Staphylococcus aureus* (MRSA) [3]. However, a large variation in AMR proportion has been reported across European countries, with a marked north-to-south gradient for most antibiotic-resistant pathogens surveyed [3]. 

AMR proportions often correlate with levels of antibiotic consumption [14]. In fact, a large variation also exists in terms of antibiotic use, as reported by data from the European Surveillance of Antimicrobial Consumption (ESAC) network [15]. However, there are also examples of countries with low antibiotic consumption but high levels of AMR, or vice versa [14,16,17]. This suggests that other factors, such as healthcare system quality, sanitation, and infection prevention and control (IPC) measures, may also play a role in the development of AMR [16,18,19,20]. For this reason, there has been growing interest in recent years in evaluating other factors that may contribute to the spread of AMR. Some of these factors include changes in the environment, such as temperature [21,22,23,24,25], as well as the exchange of genetic material between bacteria, which can lead to the spread of AMR genes [26,27,28]. Additionally, studies have also shown that population density, poor governance, reduced public health-care expenditure, abuse of disinfectants and biocides, and the presence of other stressors in the environment can also play a role in the development and spread of AMR [14,18,26,27,28,29,30,31]. Understanding the interplay between these factors is crucial for developing effective strategies to prevent and control the spread of antibiotic-resistant bacteria [29,32]. 

In the past decade, temperature changes in European countries have been observed and reported, resulting from human activities such as industrialization and deforestation, and from natural factors such as solar radiation and ocean currents [33]. This warming trend is expected to continue, and its effects on AMR are the subject of ongoing research. Previous studies have suggested a relationship between temperature change and AMR [21,22,23,24,25]. For example, some studies have found that higher temperatures can increase the rate at which bacteria develop resistance to antibiotics [34,35,36,37]. This is because warm temperatures sustain bacterial reproduction, which in turn increase the frequency of genetic mutations and the likelihood of AMR [38,39]. At the population level, two independent studies recently used observational data to analyze the impact of local temperature on the spread of antibiotic-resistant bacteria across European countries [23,24]. Their results showed that AMR rates may increase with increasing temperatures in some countries, suggesting a link between climate warming and the spread of antibiotic-resistant bacteria [23,24]. However, it is unclear whether these findings support a cause-and-effect relationship between temperature and AMR or if they simply reflect a geographical gradient. In fact, other factors, such as the use of antibiotics and population density, also played a role in the development and spread of AMR [23,24].

To better investigate this topic, we tested the hypothesis that countries that experienced the greatest temperature change over the past decade have also reported a higher increase in AMR rates. With this purpose, we carried out a ten-year ecological analysis to evaluate whether temperature changes, and not simply local temperature, may have been associated with AMR proportions across European countries and over the years 2010–2019. The analysis also included important predictors and/or confounders that can determine the geographical gradient observed, helping to explain the hypothetical relationship reported by previous studies.

## 2. Results

### 2.1. Trend of AMR and Temperature Changes across European Countries

Between 2010 and 2019, there was a significantly increasing trend in the weighted average of AMR in Europe (Figure 1A). However, some differences by country, pathogen, and antibiotic class should be discussed (Appendix A). AMR proportions were generally higher for *K. pneumoniae* than *E. coli* and *S. aureus*. While the proportion of *E. coli* resistant to carbapenem remained low overall, some countries exhibited an increasing trend of *K. pneumoniae* resistant to carbapenems. For all other combinations between Gram-negative pathogens and antibiotic classes, a moderate increase over time was observed for most countries, especially up until around 2014. For *S. aureus*, instead, there was a generally decreasing trend in the percentage of MRSA isolates. Except for carbapenem resistance, there were large variations between countries for all pathogens and antibiotic classes under consideration. 

In the same years, there was also a significantly increasing trend in the annual temperature change (Figure 1B), even though the magnitude of the change varied across European countries (Appendix A). With some exceptions, most of countries had a mean annual temperature change 1.0 °C higher than normal over the years, and all countries recorded mean annual temperature changes above 1.0 °C in 2019. Note that both GDP per capita and the governance index were significantly and negatively associated with the annual temperature change across all countries and over the years (Appendix A).

### 2.2. Relationship between Temperature Change and AMR

Although we found no evidence of a linear association between annual temperature change and AMR across all countries, years, pathogens, and antibiotic classes (β = 0.042; 95%CI = −0.016; 0.101; *p* = 0.102; Figure 2A), some relationships were evident for specific pathogen–antibiotic combinations (Figure 2B–D). In particular, we observed a significant positive relationship between annual temperature change and AMR proportion of *E. coli* resistant to third-generation cephalosporins (β = 0.192; 95%CI = 0.014; 0.370; *p* = 0.034) and *K. pneumoniae* resistant to fluoroquinolones (β = 0.250; 95%CI = 0.071; 0.429; *p* = 0.006). Marginally non-significant evidence was observed for a positive relationship with *K. pneumoniae* resistant to third-generation cephalosporins (β = 0.172; 95%CI = −0.008; 0.351; *p* = 0.060), and a negative relationship with MRSA (β = −0.172; 95%CI = −0.350; 0.006; *p* = 0.059) and *E. coli* resistant to aminopenicillins (β = −0.219; 95%CI = −0.449; 0.012; *p* = 0.063).

We also observed differences in antibiotic consumption and population density across countries and over the years (Appendix A). Unadjusted models confirmed that these predictors had a significant positive relationship with AMR proportion across all countries, years, pathogens, and antibiotic classes (Appendix A). To compare our results with those obtained by previous studies [24,25], we therefore applied a multivariable linear regression model relating temperature change to AMR and adjusting for country, time, population density, antibiotic consumption, and the interaction between time and temperature change (Appendix A). We found evidence of a positive linear association between annual temperature change and AMR proportion across all countries, years, pathogens, and antibiotic classes (β = 0.140; 95%CI = 0.039; 0.241; *p* = 0.007). In the multivariable model, both antibiotic consumption (β = 1.163; 95%CI = 1.044; 1.282; *p* < 0.001) and population density (β = 0.165; 95%CI = 0.135; 0.196; *p* < 0.001) remained significantly and positively associated with AMR proportions.

### 2.3. Accounting for the Effect of Gross Domestic Product and Governance

However, European countries also differed in GDP per capita and governance index (Appendix A), two predictors that were negatively associated with AMR proportion in unadjusted linear models (Appendix A). We therefore developed a multivariable linear regression model that further adjusted the relationship between temperature change and AMR for GDP per capita and the governance index (Appendix A). We observed that temperature change was no longer associated with AMR proportion, and that the main predictors were antibiotic consumption (β = 0.506; 95%CI = 0.366; 0.646; *p* <0.001), population density (β = 0.143; 95%CI = 0.116; 0.170; *p* < 0.001), and the governance index (β = −1.043; 95%CI = −1.207; −0.879; *p* < 0.001). Note that the latter exhibited a negative relationship, indicating that AMR proportion decreased with the increasing governance index. A positive association with time was also evident (β = 0.051; 95%CI = 0.022; 0.081; *p* = 0.007), but not with the interaction term. 

The latter multivariable model was also applied for each combination between pathogen and antibiotic (Table 1), and no evidence of association with the annual temperature change was found. A significantly increasing trend over the years was observed for AMR proportion of *E. coli* and *K. pneumoniae* independent of the antibiotic tested. Except for *K. pneumoniae* resistant to carbapenems, the governance index was negatively associated with AMR proportion of all pathogen–antibiotic combinations. For most combinations of pathogens and antibiotics, positive relationships also remained significant, along with population density and antibiotic consumption. GDP per capita, instead, showed a relationship that was positive for *K. pneumoniae* resistant to aminoglycosides and negative for MRSA.

### 2.4. Evaluating the 10-Year Effect of Temperature Change

Finally, we analyzed whether countries experiencing greater temperature changes also demonstrated greater differences in AMR over the past decade. Maps in Figure 3 depict the 10-year average temperature change and 10-year AMR difference across European countries. Considering the 10-year average temperature change, the highest values were recorded in Slovenia (1.9 °C) and Austria (1.8 °C), and another 26 countries had a 10-year average temperature change higher than or equal to 1 °C. Only Ireland (0.8 °C) and the United Kingdom (0.9 °C) recorded values less than 1 °C. These estimates determined a west-to-east increasing gradient across European countries. When looking at the 10-year difference in AMR, Italy and Spain reported the highest values, but the other 20 countries had higher weighted AMR averages in 2019 than in 2010. These estimates determined a modest north-to-south and west-to-east increasing gradient in Europe. 

When comparing these rankings, it became clear that countries experiencing greater temperature changes did not necessarily show a greater difference in AMR between 2010 and 2019. Accordingly, no correlation was evident between the 10-year average temperature change and the 10-year AMR difference across European countries (Figure 4).

## 3. Discussion

By analyzing data from 30 European countries over a ten-year period, we attempted to shed light on the relationship between temperature change and AMR. In general, our results showed that AMR proportions increased with increasing temperatures; however, the complexity and the multifaceted nature of the relationship highlights the need for understanding all the potentially involved factors. Using unadjusted models, we found significant or marginally significant relationships between temperature change and AMR, including positive associations for *E. coli* resistant to third-generation cephalosporins and *K. pneumoniae* resistant to fluoroquinolones or third-generation cephalosporins, and negative associations for *E. coli* resistant to aminopenicillins and MRSA. The findings of our study, using annual temperature change as the predictor, were partially consistent with those of previous studies which examined the association between AMR and annual minimum temperatures in the US and Europe [24,25]. 

Therefore, we replicated the model applied by McGough and colleagues [24] and adjusted for factors contributing to AMR (i.e., antibiotic consumption and population density) in order to study the relationship more deeply. In fact, antibiotic consumption and population density are important contributors to AMR and can explain the observed variations between countries. Specifically, multivariable analysis showed a positive linear association between annual temperature change and AMR proportion across all countries, years, pathogens, and antibiotic classes. These findings generally supported the associations seen in previous studies. A study in Europe, despite not examining the impact of temperature on AMR, found that increasing distance from the equator was a predictor of the prevalence of Acinetobacter spp. resistant to carbapenems [40]. A cross-sectional analysis of 30 European countries added to this evidence, reporting that mean temperature in the warm season was a significant predictor of the six-year prevalence of carbapenem-resistant *K. pneumoniae*, multi-resistant *E. coli*, and MRSA [23]. Similar results were obtained from the ecological study of McGough and colleagues, which is probably the most comprehensive on the relationship between temperature and AMR in Europe [24]. The authors found that warmer ambient temperature—assessed through the annual minimum temperature over a 17-year period (2000–2016)—was associated with a higher percentage of resistance for *E. coli* and *K. pneumoniae* [24]. There was also some research looking at the potential relationship between temperature and AMR outside of Europe, in particular in China and the US [21,25]. According to Li and colleagues, the prevalence of *K. pneumoniae* and *Pseudomonas aeruginosa* resistant to carbapenems increased with increasing average ambient temperature in 28 Chinese provinces over a 15-year period (2005–2019) [21]. Across the US, MacFadden and colleagues reported an association between an increase in ambient temperature and increases in AMR of *E. coli*, *K. pneumoniae*, and *S. aureus* between 2013 and 2015 [25]. 

Our results from the multivariable analysis also revealed positive associations between AMR and both antibiotic consumption and population density, which align with prior findings in the US and China [21,25]. However, it is necessary to note that McGough and colleagues reported a negative association between population density and AMR across European countries [24], indicating a need to assess the reliability of this relationship more thoroughly. Despite the limited understanding of the mechanisms by which temperature can impact AMR, it plays a role in various interconnected domains, including agriculture, animals, humans, and the environment as a whole [41]. First of all, the antimicrobial activity of certain antibiotics could be influenced by the temperature, either by enhancing their uptake or increasing rates of chemical catalysis [35]. From a biological perspective, bacteria can exist in the environment, as well as in animals and humans, where they may exhibit intrinsic or acquired mechanisms of resistance against antibiotics [35]. These mechanisms can be transmitted vertically through strain replication or horizontally from other bacteria, viruses, and the environment [35,36]. The impact of temperature on the acquisition and transmission of AMR is a matter of concern: on the one hand, warmer temperatures may accelerate bacterial growth, facilitating overall transmission and selection through antibiotic pressure [34]; on the other hand, the temperature may influence horizontal gene transfer, including that relating to antibiotic resistance genes [39]. It is also worth mentioning that the warming of the climate may increase the carriage of antibiotic-resistant pathogens in humans and animals [42], as well as the diversity of antibiotic resistance genes in the environment [37]. 

To gain a clearer understanding of whether the rise in AMR rates is a result of temperature change or just the convergence of different geographical gradients due to other factors, we further refined our analysis by considering GDP per capita and the governance index. The first is a summary measure of the country’s economic output per person, which was previously associated with climate change and AMR [17,43,44,45]. As a general trend, there is a correlation between GDP per capita and carbon dioxide (CO_2_) emissions, where an increase in GDP per capita results in a corresponding rise in CO_2_ emissions [46]. Instead, there is a controversy surrounding the relationship between temperature and GDP per capita, with the correlation differing based on the specific country or region and the various factors that impact both [47,48]. In certain scenarios, it has been observed that elevated temperatures can result in decreased economic productivity and a decrease in GDP per capita [48]. Moreover, the relationship between GDP per capita and AMR is complex and not well understood. However, some studies suggested that there was a correlation between a country’s wealth and the prevalence of AMR: higher GDP per capita tended to be associated with greater use of antibiotics, higher levels of industrialization and urbanization, and greater exposure to antimicrobial agents, all of which can contribute to the development and spread of AMR [17,45]. On the other hand, higher income can also increase access to better healthcare and improved sanitation, which can reduce the spread of AMR [17,45]. To consider the impact of effective and accountable management in a country, we also created an aggregate governance index using the six WGI provided by the World Bank [49]. Effective governance—characterized by transparent and accountable systems, strong regulations and enforcement, and functioning institutions—can play a key role in mitigating and adapting to the effects of climate change and in addressing the drivers of AMR [14,16,18,30,50,51]. In particular, good governance can support the development and implementation of policies and programs that promote the rational use of antibiotics, improve IPC measures, and enhance surveillance and monitoring of AMR [14,16,18,30]. On the contrary, poor governance—characterized by weak regulatory systems, inadequate enforcement, and limited accountability—can lead to a lack of action or ineffective actions against climate change and AMR [14,16,18,30]. Furthermore, weak governance can contribute to the improper use of antibiotics and hinder the resources and ability needed to address AMR effectively, including investment in research and development of new antibiotics and alternative treatments [14,16,18,30]. 

When these factors were included in the fully-adjusted multivariable model, the relationship between temperature and AMR was no longer significant, both overall and stratified by pathogen and antibiotic class. These results were consistent with those reported by Collignon and colleagues, who evaluated anthropological and socioeconomic factors contributing to AMR globally [18]. The authors showed a positive correlation between temperature and AMR in the bivariate analysis. However, temperature was no longer associated with AMR in their multivariable analysis, which adjusted for antibiotic use, GDP per capita, and indexes of governance, health expenditure, education, and infrastructure [18]. Therefore, although the relationship between temperature and AMR may seem interesting, it may be biased and explained by other factors. For instance, it is undeniable that some cold countries with low AMR rates are also those with better governance (e.g., Scandinavian countries), while some hot countries have worse governance and higher levels of resistance (e.g., Greece and Italy). In the fully-adjusted multivariable model, we found that governance index, but not GDP per capita, was a strong predictor of AMR proportion across all countries, years, pathogens, and antibiotic classes. This relationship was confirmed for nearly all combinations of pathogens and antibiotic classes. Regarding GDP per capita, the results were inconsistent, showing a positive association with the proportion of K. pneumoniae resistant to aminoglycosides and a negative association for K. pneumoniae resistant to third-generation cephalosporins. These results were also in line with the study by Collignon and colleagues, as the level of governance but not GDP per capita was one of the main predictors of AMR in their multivariable model [18]. To corroborate our findings, we evaluated the hypothesis that countries reporting the largest changes in temperature over ten years were also those that showed the greatest increase in AMR proportion. Instead, some of the countries reporting the highest temperature change (e.g., Slovenia, Austria, and Hungary) were at the bottom of the ranking for change in AMR proportion. On the other hand, countries reporting the largest increase in AMR proportion did not necessarily experience a high increase in temperature. 

Our study and analysis shared some limitations with previous works on the same issue [21,23,24,25]. Firstly, due to the ecological nature of the study, we were unable to infer the causality of the relationships observed. Based on previous studies and biological plausibility, we selected a set of predictors that may intervene in the relationship between temperature and AMR. However, the effect of residual confounding cannot be completely excluded, especially that attributable to unmeasured factors (e.g., infrastructures, education, healthcare access, sanitation, policies on antibiotic consumption and AMR, and antibiotic consumption in hospitals and in the animal sector). Secondly, there were gaps in national data reporting over time, with some systematic and/or random missing data for AMR proportions and antibiotic consumption. To compile the dataset, we used the more comprehensive data sources for AMR (EARS-Net) and antibiotic consumption (ESAC-Net) in Europe. Nonetheless, it is worth mentioning that AMR proportions were restricted to bacterial isolates from invasive bloodstream infections and cerebrospinal fluid samples, and that antibiotic consumption data referred to the overall consumption of antibiotics for systemic use (ATC group J01) in the community. We recognize that it would have been more appropriate to include the consumption of specific classes of antibiotics, but the data was not complete for all countries and years considered. Thirdly, all the data were available at the country and year level. Therefore, evaluating the observed relationships by smaller geographic areas or by seasons was not possible.

Compared to previous studies, our analysis presents novelties that also represent some of its strengths. Firstly, we evaluated temperature as the mean temperature change with respect to a baseline climatology and not simply as the annual average temperature. Secondly, we used a larger amount of information on AMR, including data on ten combinations of pathogens and antibiotics. The period covered by our analysis was also consistent with previous studies. Finally, the introduction of additional predictors, such as the governance index, allowed us to disentangle the relationship between temperature and AMR.

Drawing inspiration from previous research, our study was designed to provide a better understanding of the relationship between temperature change and AMR. Although warmer countries may report higher proportions of AMR, our study did not find a significant effect of temperature change. Rather, other factors—such as antibiotic use and a country’s governance—seem to determine a north-to-south geographical gradient for AMR. As a result of this evidence, ensuring the appropriate use of antibiotics and improving governance efficiency are the most effective ways to counteract AMR. It is necessary to conduct further experimental studies and obtain more detailed data to investigate whether and how climate change affects AMR.

## 4. Materials and Methods

### 4.1. Study Design

We conducted an ecological study of the evolution of AMR proportion of three common bacterial pathogens and different antibiotic classes over a 10-year period (2010–2019) across European countries and evaluated associations with temperature changes and other predictors. Data from 2020 onwards were not used to prevent the analysis from being affected by the COVID-19 pandemic. The study sample consisted of the following 30 countries from the European Union (EU) or the European Economic Area (EEA) and those participating in the EARS-net surveillance network [6]: Austria, Belgium, Bulgaria, Croatia, Cyprus, Czechia, Denmark, Estonia, Finland, France, Germany, Greece, Hungary, Iceland, Ireland, Italy, Latvia, Lithuania, Luxembourg, Malta, Netherlands, Norway, Poland, Portugal, Romania, Slovakia, Slovenia, Spain, Sweden, and United Kingdom.

### 4.2. Antimicrobial Resistance

We used data on AMR proportions of three pathogens (i.e., *E. coli*, *K. pneumoniae*, and *S. aureus*) that are the most common Gram-positive and Gram-negative pathogens causing infections globally. This choice was also motivated to align with previous published studies on the same issue [24,25]. All data were collected at the country level as part of national surveillance, and gathered in the EARS-Net database [52]. This is a publicly available database including AMR data of the most common combinations between pathogens and antibiotics for most of the European countries since 2000. 

In this study, we analyzed data of 30 European countries for the following combinations between pathogens and antibiotics: *K. pneumoniae* resistant to carbapenems, fluoroquinolones, third generation cephalosporins, and aminoglycosides; *E. coli* resistant to aminopenicillins, fluoroquinolones, and third generation cephalosporins; and MRSA. AMR proportion (%) was defined as the percentage of tested bacterial isolates that were non-susceptible to a given antibiotic for a given year, pathogen, and antibiotic class. To produce a summary measure of AMR, we first calculated the weighted average of AMR proportions for each country and year, normalized by pathogen and antibiotic class and weighted by the number of isolates tested [24,25]. We next computed the 10-year AMR difference as the difference of weighted AMR averages between 2019 and 2010.

### 4.3. Temperature Change

Data on the mean annual temperature change were obtained from the FAOSTAT database, which provides information on surface air temperature changes (°C) for 192 countries over the period 1961–2020 [53]. Country-level temperature statistics are produced in collaboration with the NASA Goddard Institute for Space Studies (NASA–GISS), and made available as monthly, seasonal, and annual mean temperature change with respect to a baseline climatology, corresponding to the period of 1951–1980. For temporal annual means, the meteorological rather than the calendar definition was used (i.e., from December 1st of the previous calendar year to November 30th of the same calendar year). In this study, we used data on the annual temperature change for 30 European countries from 2010 to 2019 [33]. To produce a summary measure of temperature change, we also computed the 10-year average temperature change for each country as the mean of annual temperature changes from 2010 to 2019.

### 4.4. Antibiotic Consumption

Antibiotic consumption data were obtained from the European Surveillance of Antimicrobial Consumption Network (ESAC-Net) database [54]. In general, these data are reported by each country through The European Surveillance System (TESSy) and in accordance with the ESAC-Net reporting protocol [55]. The ESAC-Net database includes data and trends for the EU/EEA and individual countries as reported for both the community and hospital sectors. Antibiotic consumption is expressed as the number of Defined Daily Doses (DDDs) per 1000 inhabitants and per day, and the Anatomical Therapeutic Chemical (ATC) classification system is used for the allocation of antimicrobials in groups [55]. In the current study, we used data on the consumption of antibiotics for systemic use (ATC group J01) in the community from 2000 to 2019. To account for differences in antibiotic consumption between countries and years, we used annual country-level antibiotic consumption data.

### 4.5. Other Predictors

In addition to temperature change and antibiotic consumption, we selected additional factors and/or confounders that may help predict AMR based on the previous literature [21,23,24,25]. We included year and country as predictors in order to capture year- and country-level confounding effects. In particular, the inclusion of country in the models inherently adjusted for all non-time varying predictors. We also included population density (expressed as persons per km^2^) and the gross domestic product per capita (GDP; expressed as current USD) for each country and year, using annual population estimates from the World Bank [56]. To include the potential effect of governance, we used Worldwide Governance Indicators (WGI) provided by the World Bank, which covered the following six dimensions: Voice and Accountability, Political Stability and Absence of Violence/Terrorism, Government Effectiveness, Regulatory Quality, Rule of Law, and Control of Corruption [56]. These indicators are based on several hundred individual underlying variables, taken from different existing data sources (e.g., surveys of firms and households, subjective assessments of a variety of commercial business information providers, non-governmental organizations, multilateral organizations, and other public-sector bodies) [49]. Each indicator is reported in its standard normal unit, with higher values indicating better outcomes. To obtain an aggregate index of governance, we computed the average of standardized values of six indicators composing the index [49].

### 4.6. Statistical Analysis

To easily compare AMR proportions across different pathogen–antibiotic combinations, we first centered the data about the mean and normalized by the standard deviation for all pathogens and antibiotic classes. The weighted annual mean of these values was then used to depict the temporal trend of AMR from 2010 to 2019. Likewise, we plotted the temporal trend of the mean annual temperature change over the ten years. 

We next plotted the relationship between mean annual temperature change and AMR for all pathogens and antibiotics, and used both simple and multivariable linear models to assess their association. In the unadjusted model, we evaluated the association between mean annual temperature change (°C) and normalized AMR values for each country, year, pathogen, and antibiotic class. In the multivariable model 1, we adjusted the relationship for country, year, population density (persons/km^2^), antibiotic consumption (DDD per 1000 inhabitants per day), and the interaction between year and temperature change. These predictors were selected to align with the study by McGough and colleagues [24] and to make results comparable. The interaction term was included to assess the extent to which temperature change was associated with AMR over the years. In the multivariable model 2, further adjustments were made for GDP per capita and the Governance index as indicators of the economic and governance spheres [14,18,30], respectively. In all models, natural log transformation was applied to antibiotic consumption, population density, and GDP per capita to improve linear fit. Multivariable model 2 was replicated across each pathogen and antibiotic class. All the results of linear models are reported as the regression coefficient with the 95% Confidence Interval (95%CI). 

In order to further understand whether countries with a greater temperature change over the years have also shown a higher change in AMR, we mapped and plotted the ranking of countries for the 10-year average temperature change and 10-year AMR difference, respectively. Spearman’s rank correlation coefficient was used to measure the rank correlation between them.

## 5. Conclusions

Although previous studies suggested a link between climate change and the spread of antibiotic-resistant bacteria—with the potential for higher AMR rates as temperatures rise—it remains uncertain whether the relationship between temperature and AMR is merely due to coincidental geographical gradients influenced by other unmeasured factors. Our ecological analysis focused on temperature change, and not simply on mean local temperature, to evaluate its association with AMR proportions across European countries and over a ten-year period (2010–2019). Compared to previous studies, we used a larger amount of information on AMR, including data on ten combinations between three pathogens and six antibiotic classes. To help unravel the relationship between temperature and AMR, our analysis also included additional predictors that had not been previously considered, such as the GDP per capita and a governance index. In line with previous findings, we initially demonstrated a positive linear association between temperature change and AMR proportion, adjusting for the effect of country, time, population density, antibiotic consumption, and the interaction between time and temperature change. However, when GDP per capita and the governance index were included in the multivariable model, temperature change was no longer associated with AMR. Accordingly, no correlation was evident when considering ten-year changes in temperature and AMR, with countries experiencing greater temperature changes that did not necessarily show a greater change in AMR. Antibiotic consumption, population density, and the governance index remained the main factors contributing to AMR. While AMR proportions increased with increasing antibiotic consumption and population density, the governance had an opposite effect. In fact, AMR proportions decreased with an increasing governance index. For this reason, ensuring the appropriate use of antibiotics and improving governance efficiency are still the most effective ways to counteract AMR. It is necessary to conduct further experimental studies and obtain more detailed data to investigate whether climate change affects AMR.

## Figures and Tables

**Figure 1 antibiotics-12-00777-f001:**
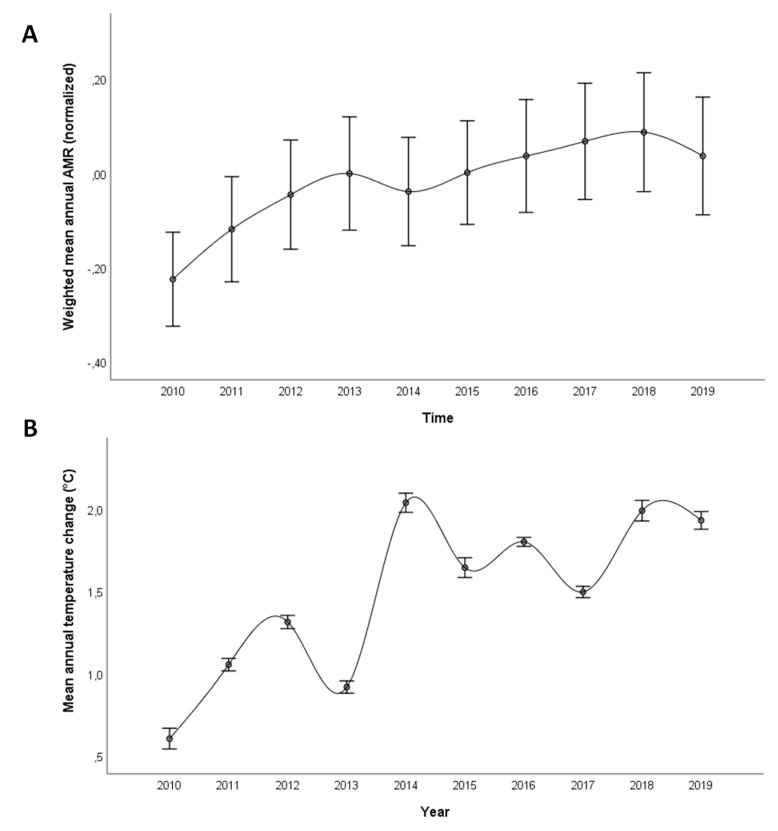
Trends of antimicrobial resistance and temperature change. (**A**) Ten-year trend of weighted average antimicrobial resistance (normalized) across European countries (2010–2019); antimicrobial resistance proportions of three pathogens (i.e., *E. coli*, *K. pneumoniae*, and *S. aureus*) were obtained from the EARS-Net database. A summary measure of antimicrobial resistance was computed as the weighted average of antimicrobial resistance proportions for each country and year, normalized by pathogen and antibiotic class and weighted by the number of isolates tested; (**B**) 10-year trend of average temperature change (°C) across European countries (2010–2019); mean annual temperature changes were obtained from the FAOSTAT database.

**Figure 2 antibiotics-12-00777-f002:**
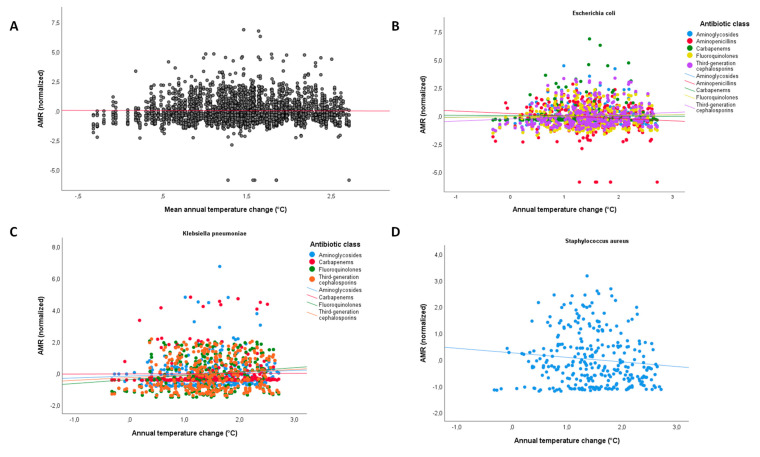
The relationship between annual temperature change and antimicrobial resistance. (**A**) Scatter plot of annual temperature change (°C) and antibiotic resistance (normalized) across all countries, years, pathogens, and antibiotic classes. The unadjusted linear trend line is shown in red; (**B**) scatter plot of annual temperature change (°C) and antibiotic resistance (normalized) of *E. coli* across all countries and years. Unadjusted linear trend lines are colored by antibiotic class; (**C**) scatter plot of annual temperature change (°C) and antibiotic resistance (normalized) of *K. pneumoniae* across all countries and years. Unadjusted linear trend lines are colored by antibiotic class; (**D**) scatter plot of annual temperature change (°C) and antibiotic resistance (normalized) of *S. aureus* across all countries and years. The unadjusted linear trend line is shown in blue.

**Figure 3 antibiotics-12-00777-f003:**
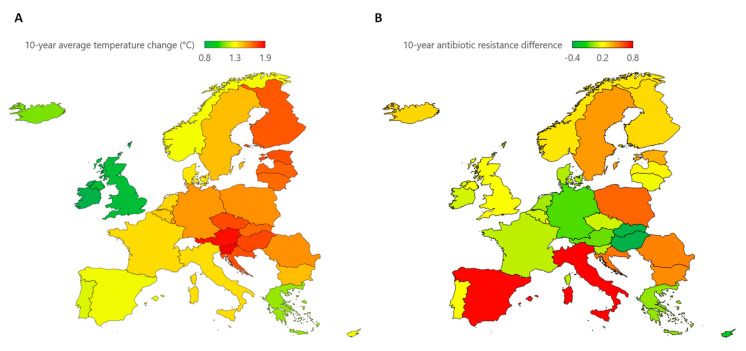
Ten-year temperature and antimicrobial resistance changes. (**A**) Map of the 10-year average temperature change (°C) across European countries; this summary measure was computed as the mean of annual temperature changes from 2010 to 2019; (**B**) map of the 10-year antimicrobial resistance difference across European countries (10-year antimicrobial resistance difference was computed as the difference of weighted averages between 2019 and 2010).

**Figure 4 antibiotics-12-00777-f004:**
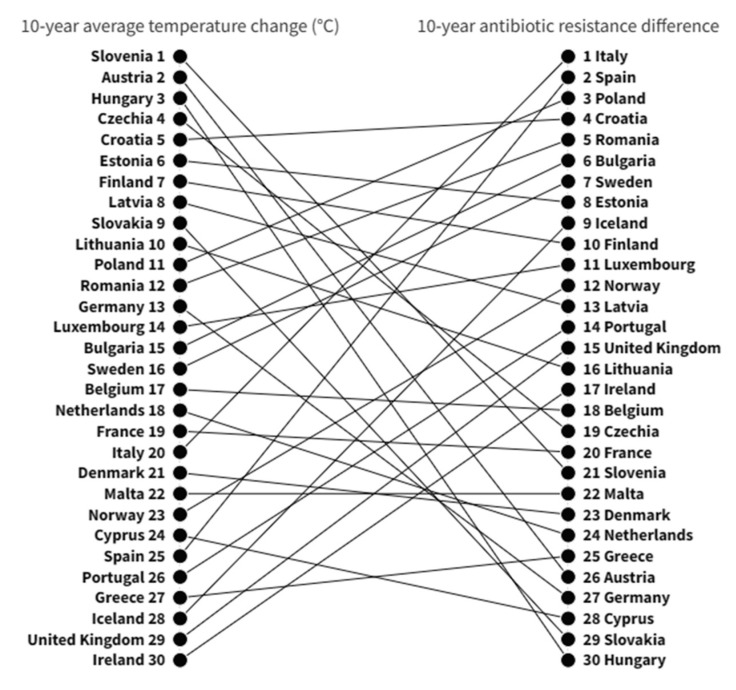
Ranking of European countries according to 10-year average temperature change and antimicrobial resistance difference. The 10-year average temperature change (°C) was computed across European countries as the mean of annual temperature changes from 2010 to 2019. The 10-year antimicrobial resistance difference across European countries was computed as the difference of weighted averages between 2019 and 2010.

**Table 1 antibiotics-12-00777-t001:** Unadjusted and adjusted multivariable analyses to evaluate the linear relationship of antimicrobial resistance with temperature change and other predictors, by pathogen and antibiotic class.

Pathogens and Predictors	Coefficient (95% Confidence Interval) ^a^						
*E. coli*	Overall	Aminoglycosides	Aminopenicillins	Carbapenems	Fluoroquinolones	Third-Generation Cephalosporins	Methicillin
Mean annual temperature change (°C)	0.034 (−0.087; 0.155)	0.168 (−0.055; 0.390)	−0.250 (−0.556; 0.057)	0.068 (−0.230; 0.365)	0.116 (−0.097; 0.329)	0.070 (−0.163; 0.304)	-
Year	0.059 (0.020; 0.098) **	0.071 (−0.001; 0.142)	0.034 (−0.064; 0.133)	0.062 (−0.033; 0.158)	0.060 (−0.008; 0.129)	0.066 (−0.009; 0.141)	-
Interaction	−0.024 (−0.048; −0.001) *	−0.047 (−0.090; −0.003) *	−0.003 (−0.063; 0.057)	−0.031 (−0.089; 0.027)	−0.028 (−0.069; 0.014)	−0.013 (−0.058; 0.033)	-
Antibiotic consumption in the community (DDD/1000 inhabitants/day)	0.654 (0.469; 0.840) ***	0.335 (−0.005; 0.676)	1.210 (0.741; 1.679) ***	0.578 (0.123; 1.033) *	0.729 (0.403; 1.054) ***	0.419 (0.061; 0.777) *	-
Population density (persons/km^2^)	0.147 (0.111; 0.183) ***	0.099 (0.033; 0.165) **	0.328 (0.236; 0.419) ***	−0.041 (−0.130; 0.047)	0.279 (0.215; 0.342) ***	0.073 (0.003; 0.142) *	-
GDP per capita (current USD)	0.148 (−0.005; 0.301)	0.147 (−0.134; 0.429)	−0.023 (−0.411; 0.365)	0.255 (−0.122; 0.631)	0.262 (−0.008; 0.531)	0.100 (−0.196; 0.396)	-
Governance Index	−1.291 (−1.508; −1.073) ***	−1.559 (−1.959; −1.158) ***	−1.127 (−1.678; −0.575) ***	−0.982 (−1.517; −0.447) ***	−1.378 (−1.761; −0.996) ***	−1.406 (−1.827; −0.985) ***	-
** *K. Pneumoniae* **							-
Mean annual temperature change (°C)	0.006 (−0.149; 0.161)	−0.071 (−0.379; 0.237)	-	−0.084 (−0.410; 0.241)	0.101 (−0.083; 0.285)	0.078 (−0.085; 0.242)	-
Year	0.051 (0.001; 0.101) *	0.167 (0.068; 0.266) **	-	0.049 (−0.056; 0.154)	−0.010 (−0.069; 0.049)	−0.002 (−0.055; 0.051)	-
Interaction	−0.006 (−0.036; 0.024)	−0.035 (−0.095; 0.026)	-	−0.003 (−0.067; 0.061)	0.012 (−0.024; 0.048)	0.001 (−0.031; 0.033)	-
Antibiotic consumption in the community (DDD/1000 inhabitants/day)	0.097 (−0.140; 0.334)	−0.479 (−0.950; 0.007)	-	−0.225 (−0.724; 0.274)	0.531 (0.250; 0.813) ***	0.560 (0.309; 0.811) ***	-
Population density (persons/km^2^)	0.111 (0.065; 0.157) ***	0.195 (0.103; 0.287) ***	-	0.138 (0.041; 0.234) **	0.075 (0.020; 0.129) **	0.036 (−0.013; 0.085)	-
GDP per capita (current USD)	0.086 (−0.110; 0.282)	0.652 (0.262; 1.042) **	-	0.162 (−0.251; 0.574)	−0.187 (−0.420; 0.046)	−0.284 (−0.492; −0.076) **	-
Governance Index	−0.804 (−1.083; −0.525) ***	−0.657 (−1.212; −0.103) ***	-	0.075 (−0.512; 0.661)	−1.328 (−1.659; −0.997) ***	−1.304 (−1.599; −1.009) ***	-
** *S. aureus* **							
Mean annual temperature change (°C)	-	-	-	-	-	-	0.075 (−0.122; 0.272)
Year	-	-	-	-	-	-	0.015 (−0.048; 0.078)
Interaction	-	-	-	-	-	-	−0.029 (−0.067; 0.010)
Antibiotic consumption in the community (DDD/1000 inhabitants/day)	-	-	-	-	-	-	1.401 (1.100; 1.702) ***
Population density (persons/km^2^)	-	-	-	-	-	-	0.251 (0.193; 0.310) ***
GDP per capita (current USD)	-	-	-	-	-	-	−0.149 (−0.398; 0.100)
Governance Index	-	-	-	-	-	-	−0.765 (−1.119; −0.412) ***

Abbreviations: DDD, defined daily dose; GDP, gross domestic product. All available combinations of three pathogens (*E. coli*, *K. pneumoniae*, and *S. aureus*) and six antibiotic classes (aminoglycosides, aminopenicillins, carbapenems, fluoroquinolones, third-generation cephalosporins, and methicillin) were analyzed. For interpretability, year was zeroed at baseline (2010) and a natural log transform was applied to antibiotic consumption, GDP per capita, and population density to improve linear fit. ^a^ Coefficients with 95% confidence intervals were adjusted for country, mean annual temperature change, year, antibiotic consumption, population density, GDP per capita, governance index, and the interaction between year and temperature change. *** *p* < 0.001; ** *p* < 0.01; * *p* < 0.05.

## Data Availability

Data are available on reasonable request from the corresponding author.

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
