# Peer review of "Socioeconomic and Governance Factors Disentangle the Relationship between Temperature and Antimicrobial Resistance: A 10-Year Ecological Analysis of European Countries"

_antibiotics, 2023, doi:10.3390/antibiotics12040777_

Round 1

Reviewer 1 Report

The manuscript, “Socioeconomic and governance factors disentangle the relationship between temperature and antimicrobial resistance: a 10-year ecological analysis of European countries” examines the association of increasing temperature change with antimicrobial resistance (AMR) across 30 European countries.  

Consistent with previously published data, the results at first demonstrate a positive correlation between AMR and increasing temperatures. However, when a multivariable model is applied that adjusts for other factors (such as antibiotic consumption and population density) no evidence of association between AMR and average annual temperature change is apparent. This discrepancy may explain the observed variations of AMR among European countries. The authors note that the increase in in AMR is multifaceted and influenced by a combination of factors; appropriate use of antibiotics and sound governance are highlighted as two key predictors of AMR resistance. 

Overall, my suggestions for improvement are minor. Something to consider (but not entirely necessary) involves lines 258-260 of the manuscript. This part describes the effect of increased temperature on various aspects of AMR. It may be an interesting point to make that antimicrobial activity of certain antibiotics is also dependent on temperature (either by enhancing uptake of antibiotics or increasing rates of chemical catalysis). This could be worth mentioning given the multifactorial nature of AMR resistance as it relates to temperature. 

Other relatively minor edits to make:

Names for organisms should be italicized. 

The caption for Figure 1 incorrectly indicates the order of graphs; A describes AMR while B describes temperature change (not the other way around). Also, numbers less than one should have a placeholder zero (e.g. 0.5 instead of .5).

Author Response

Dear Editor,

please consider the revised version of the manuscript entitled “Socioeconomic and governance factors disentangle the relationship between temperature and antimicrobial resistance: a 10-year ecological analysis of European countries” in which we have addressed all comments and suggestions from the Reviewers. This letter is intended for the convenience of the Editor and Reviewers and contains the list of the requested changes. The following list of changes and answers to comments of Reviewers addresses all revisions made in the manuscript (in red font).

Reviewer 1: The manuscript, “Socioeconomic and governance factors disentangle the relationship between temperature and antimicrobial resistance: a 10-year ecological analysis of European countries” examines the association of increasing temperature change with antimicrobial resistance (AMR) across 30 European countries.  Consistent with previously published data, the results at first demonstrate a positive correlation between AMR and increasing temperatures. However, when a multivariable model is applied that adjusts for other factors (such as antibiotic consumption and population density) no evidence of association between AMR and average annual temperature change is apparent. This discrepancy may explain the observed variations of AMR among European countries. The authors note that the increase in in AMR is multifaceted and influenced by a combination of factors; appropriate use of antibiotics and sound governance are highlighted as two key predictors of AMR resistance. 

Answer: We would like to take this opportunity to thank the Reviewer for his/her positive comment on our study.

R: Overall, my suggestions for improvement are minor. Something to consider (but not entirely necessary) involves lines 258-260 of the manuscript. This part describes the effect of increased temperature on various aspects of AMR. It may be an interesting point to make that antimicrobial activity of certain antibiotics is also dependent on temperature (either by enhancing uptake of antibiotics or increasing rates of chemical catalysis). This could be worth mentioning given the multifactorial nature of AMR resistance as it relates to temperature. 

A: As suggested, we have included this point in the discussion section.

R: Names for organisms should be italicized. 

A: As suggested, we put the names of microorganisms in italics.

R: The caption for Figure 1 incorrectly indicates the order of graphs; A describes AMR while B describes temperature change (not the other way around). Also, numbers less than one should have a placeholder zero (e.g. 0.5 instead of .5).

A: We apologize for the mistake that has been corrected. However, the software we used does not allow to modify the format of numbers in x and y axes.

Reviewer 2 Report

1.     The manuscript addresses the prevalence of antimicrobial resistance (AMR) across European countries over the last decade. The authors have investigated how various environmental and socioeconomic factors impact AMR in these countries. For this authors studied the relationship between a wide range of factors and AMR over 10 years. 

2.     They used various data sources to interpret the relationships and found that there is a positive linear association between temperature change and 18 AMR proportion across all the countries, years, pathogens, and antibiotics. They also showed that GDP per capita and governance play a huge part in controlling the AMR. This work provides more comprehensive view on the factors which could affect the temperature changes which were lacking in previous studies.

3.     Overall experiments are well performed and presented. The references cited are relevant for the study. The conclusions made in the study are well justified. The study gives an insight into the prevalence of AMR over the years and how it can be tackled in the future. The impact of this study on advancing an understanding of antimicrobial resistance does feel strong.

4.     Minor comments

·       Name of the species should be in italics.

·       The font size of the figure/graph legends is too small.

·       Authors should modify the figure legends with more detailed experimental information instead of just what graph represents.

·       Figure 4 showing ranking of European countries according to 10-year average temperature change and antimicrobial resistance difference is not the best way to represent the data. The author should reconsider showing the ranking in different way.

Author Response

Dear Editor,

please consider the revised version of the manuscript entitled “Socioeconomic and governance factors disentangle the relationship between temperature and antimicrobial resistance: a 10-year ecological analysis of European countries” in which we have addressed all comments and suggestions from the Reviewers. This letter is intended for the convenience of the Editor and Reviewers and contains the list of the requested changes. The following list of changes and answers to comments of Reviewers addresses all revisions made in the manuscript (in red font).

Reviewer 2: The manuscript addresses the prevalence of antimicrobial resistance (AMR) across European countries over the last decade. The authors have investigated how various environmental and socioeconomic factors impact AMR in these countries. For this authors studied the relationship between a wide range of factors and AMR over 10 years. They used various data sources to interpret the relationships and found that there is a positive linear association between temperature change and 18 AMR proportion across all the countries, years, pathogens, and antibiotics. They also showed that GDP per capita and governance play a huge part in controlling the AMR. This work provides more comprehensive view on the factors which could affect the temperature changes which were lacking in previous studies. Overall experiments are well performed and presented. The references cited are relevant for the study. The conclusions made in the study are well justified. The study gives an insight into the prevalence of AMR over the years and how it can be tackled in the future. The impact of this study on advancing an understanding of antimicrobial resistance does feel strong.

Answer: We would like to take this opportunity to thank the Reviewer for his/her positive comment on our study.

R: Name of the species should be in italics.

A: As suggested, we put the names of microorganisms in italics.

R: The font size of the figure/graph legends is too small.

A: As suggested, we have increased the font size of the text in figure legends.

R: Authors should modify the figure legends with more detailed experimental information instead of just what graph represents.

A: As indicated in the instructions for authors, figure legends should describe what is contained in the figures. Anyway, as requested, we have added more details for each figure.

Reviewer 3 Report

The manuscript is good in shape. I have very few comments that should be addressed before further consideration.

Discuss the current situation of the AMR problem in European counties where the AMR rate is significantly higher.

The conclusion section needs revisions. It should be as precise as possible. 

Author Response

Dear Editor,

please consider the revised version of the manuscript entitled “Socioeconomic and governance factors disentangle the relationship between temperature and antimicrobial resistance: a 10-year ecological analysis of European countries” in which we have addressed all comments and suggestions from the Reviewers. This letter is intended for the convenience of the Editor and Reviewers and contains the list of the requested changes. The following list of changes and answers to comments of Reviewers addresses all revisions made in the manuscript (in red font).

Reviewer 3: The manuscript is good in shape. I have very few comments that should be addressed before further consideration.

Answer: We would like to take this opportunity to thank the Reviewer for his/her positive comment on our study.

R: Discuss the current situation of the AMR problem in European counties where the AMR rate is significantly higher.

A: As suggested, in the introduction section, we have added more details on the AMR situation of European countries.

R: The conclusion section needs revisions. It should be as precise as possible. 

A: As suggested, we have revised the conclusion section.